# Echoes of the Visual Past: Test-Time Prompt Tuning with Multi-Scale Visual Memory

## Abstract

Test-time prompt tuning (TPT) aims to adapt pre-trained vision-language models (VLMs) to various downstream tasks by learning textual prompts using unlabeled data at test time. However, existing TPT methods exhibit a performance gap compared to a line of prompt-engineering-based methods that leverage hand-crafted or LLM-generated prompts for VLM adaptation. We attribute this gap to a core limitation of previous TPT approaches: they learn prompts from only limited class-specific visual knowledge derived from a single test image. As a result, the learned prompts underperform compared to hand-crafted and LLM-generated prompts enriched with diverse, class-specific knowledge. To address this limitation, we propose **T**est-time **P**rompt **T**uning with **M**ulti-scale visual **M**emory ($M^2$TPT). Specifically, the memory is constructed to store past seen class-relevant image patches as multi-scale visual descriptions for each class. For each test image, we use it to query the memory and learn the textual prompt using both the test image and the retrieved class-relevant visual memory. Additionally, we introduce holistic visual memory to better handle holistic visual recognition tasks that require global image-level context, and an irrelevance suppression strategy to mitigate the impact of noisy memory entries at test time. We evaluate our method on 15 commonly used benchmark datasets and show that it outperforms existing TPT methods. Furthermore, our framework can incorporate human-designed prompts and achieves state-of-the-art performance compared to recent VLM adaptation methods that use hand-crafted or LLM-generated prompts.

## 1 Introduction

Pre-trained vision-language models (VLMs) have demonstrated powerful representational capabilities, making them valuable for a wide range of computer vision tasks [35, 16, 23, 24, 25]. To efficiently adapt VLMs to downstream tasks and new domains, the prompt-tuning paradigm has been explored—where only the input text context is optimized using limited test data, while the model backbone remains frozen [48, 47]. More practically, recent research has developed test-time prompt tuning (TPT), which directly optimizes prompts using unlabeled test data streams [38].

Aside from prompt-tuning-based methods, a line of prompt-engineering-based methods have designed hand-crafted and LLM-generated prompts tailored for each dataset to adapt VLMs to target tasks [34, 18, 46, 50]. Recently these methods have significantly outperformed prompt-tuning approaches on image classification benchmarks, as illustrated in Fig. 1a. Human-designed prompts introduce prior dataset knowledge and rich class-specific information, making them more effective than prompts learned from a generic "a photo of a [CLASS]" initialization during test time. The red dashed lines show the performance of these methods when using a generic prompt, highlighting that the performance gap mainly lies between the learned prompt and the human-designed prompts. However,

Submitted to 39th Conference on Neural Information Processing Systems (NeurIPS 2025). Do not distribute.

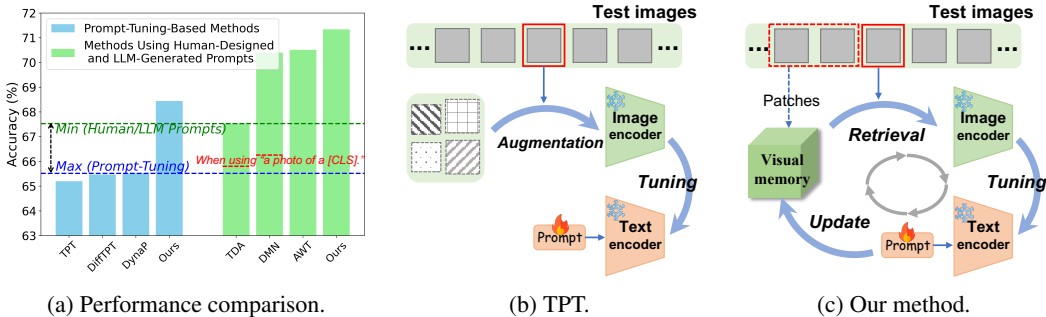

(a) Performance comparison.  (b) TPT.  (c) Our method.

Figure 1: **(a) Performance comparison on 10 downstream image classification datasets.** Existing test-time prompt tuning (TPT) methods exhibit a performance gap compared to adaptation methods that use hand-crafted or LLM-generated prompts, as illustrated by the blue and green dashed lines. The red dashed lines show the performance of these methods when using a generic prompt, highlighting that the performance gap primarily lies between the TPT-learned prompt and the human-designed prompts. **(b) TPT** [38]. Previous TPT methods typically optimize a learnable prompt using only the visual information from the current test image and its augmentations. **(c) Our method** enhances test-time prompt learning with memorized past visual descriptions for each class and introduces a mutual promotion framework between the learnable prompt and the evolving visual memory.

prompt-engineering-based methods require prior knowledge of the test datasets and additional time or effort to design or generate effective prompts. In contrast, TPT methods can adapt to unlabeled test streams on the fly without relying on human intervention. Has the potential of TPT methods truly been exhausted?

As shown in Fig. 1b, prior TPT methods typically optimize a trainable prompt using only the current test image and its augmentations, relying on unsupervised losses such as entropy minimization [38] and distribution alignment [1]. We argue that such methods fail to learn prompts from sufficient class-specific visual knowledge due to their reliance on limited visual information from a current test image, which limits their competitiveness compared to human-designed prompts enriched with diverse and explicit class- and dataset-level knowledge. To address this limitation, we propose test-time prompt tuning enhanced by a past visual memory containing class-specific visual descriptions.

In our approach, as depicted in Fig. 1c, we construct a multi-scale visual memory by accumulating visual patches that are highly relevant to each class at every time step during the test stream. Before prompt tuning, the current test image is used as a query to retrieve semantically related visual patches from this memory. The textual prompt is then optimized using both the test image and the retrieved, diverse, class-relevant visual information from the same test distribution, enabling the prompt tuning process to more effectively capture class-specific knowledge. Reciprocally, the visual memory also benefits from the learned prompt, as it is updated based on the optimized prompt. The three sequential steps—memory retrieval, prompt tuning, and memory update—achieve a round of mutual promotion between the tunable textual prompt and the evolving visual memory for each test image.

In addition to object recognition, downstream tasks may require holistic visual understanding, such as scene understanding [41] and land cover classification [13], which demand comprehensive image-level context that may be lost when focusing solely on patches. To this end, we further construct a holistic visual memory that retains class-relevant full-view images and functions in coordination with the multi-scale memory. Moreover, because memory update and retrieval operate without ground-truth supervision at test time, the visual memory can inevitably be noisy. To mitigate adverse effects, we introduce an irrelevance suppression strategy: we filter out low-relevance memory entries from the retrieved class-specific memory during retrieval, and we maintain a class-irrelevant memory that stores previously seen misleading patches from the test domain. This irrelevant memory is used to penalize high-confidence but incorrect cues during prompt tuning, thereby suppressing distracting and misleading information.

We evaluate our test-time prompt tuning method on 15 datasets, including commonly used downstream image classification benchmarks and out-of-distribution datasets. Our method outperforms existing test-time prompt tuning methods without prompt engineering. Furthermore, our framework can also

benefit from human-designed prompts, enabling it to achieve state-of-the-art performance compared to recent VLM adaptation methods that rely on hand-crafted or LLM-generated prompts.

## 2 Related work

**Prompt learning.** As vision-language models (VLMs) have demonstrated strong performance across various computer vision tasks, recent research has explored prompt learning as a parameter-efficient approach to adapt VLMs to real-world downstream scenarios [27, 12, 5, 22, 17, 20]. CoOp [48] proposes learning a contextual prompt in the input space of the text encoder using few-shot data, while keeping the model backbone frozen. CoCoOp [47] improves upon CoOp by introducing condition tokens derived from input images into the textual prompt learning process, enabling better generalization. In contrast, Bahng et al. [3] introduce visual prompt learning, which operates on the image encoder of VLMs. MaPLe [19] further advances this line of work by jointly learning prompts on both the image and text encoders to enhance transfer learning performance.

**Test-time prompt tuning.** To improve the generalization ability of VLMs without requiring labeled test data, TPT [38] proposes test-time prompt tuning (TPT). This pioneering method learns adaptive textual prompts from the current test image and its augmentations using an entropy minimization objective, while keeping the model backbone frozen. PromptAlign [1] explicitly addresses distribution shift by introducing a distribution statistics alignment loss to guide test-time prompt optimization. C-TPT [43] considers the calibration of VLMs for prompt tuning at test time. More recently, HisTPT [44] and DynaPrompt [42] propose online test-time prompt tuning methods to leverage past information during inference. HisTPT [44] constructs long-term and short-term knowledge banks that store output text features generated from prompts, providing self-regularization to stabilize online prompt learning. DynaPrompt [42] maintains a prompt pool containing multiple prompts and selects among them for stable online optimization. In our method, we do not follow this continuous test-time prompt tuning paradigm, but instead adopt the original setting introduced by TPT [38], in which an adaptive prompt is learned from scratch for each test sample independently.

**VLM adaptation with prompt engineering.** Apart from prompt-tuning-based methods, another line of research explores prompt-engineering-based VLM adaptation [29, 37, 34, 32, 11]. CuPL [34] leverages large language models (LLMs) [2] to generate textual descriptions for each class in the test dataset, replacing the generic prompt with these customized ones to improve prediction accuracy. TDA [18] and DMN [46] adopt hand-crafted prompts and LLM-generated prompts, respectively, on the text branch. On the vision branch, they design memory-based methods that perform non-parametric learning with visual features in a manner similar to the k-nearest neighbors (KNN) algorithm [30], to improve zero-shot classification. More recently, AWT [50] uses LLMs to generate class-specific prompt candidates and transforms the test image into multiple views, then formulates image–text matching as an optimal transport problem for zero-shot classification. While prompt-engineering-based methods have demonstrated effectiveness, they require prior knowledge of the test dataset and additional effort to craft or generate prompts. In contrast, TPT methods aim to adapt VLMs on the fly, focusing on test-time prompt learning without relying on human supervision.

## 3 Method

In this section, we first introduce the preliminaries of CLIP and test-time prompt tuning in Sec. 3.1. Then, Sec. 3.2 describes the overall framework of our method and its main component, the multi-scale visual memory. Secs. 3.3 and 3.4 present the remaining two components of our method.

### 3.1 Preliminaries

**CLIP.** The Contrastive Language–Image Pre-training (CLIP) model [35] comprises an image encoder $f(\cdot)$ and a text encoder $g(\cdot)$, which are pre-trained on large-scale image–text pairs using contrastive learning. Once pre-trained, CLIP can perform zero-shot image classification on a variety of downstream datasets. For a dataset with $C$ classes, CLIP first encodes each class using a generic prompt $\mathbf{p}$, such as "a photo of [CLASS c].", producing class-specific text embeddings $\{\mathbf{p}_c\}_{c=1}^{C}$. Given a test image $\mathbf{X}$, CLIP compares its encoded feature $f(\mathbf{X})$ with the text embeddings of all

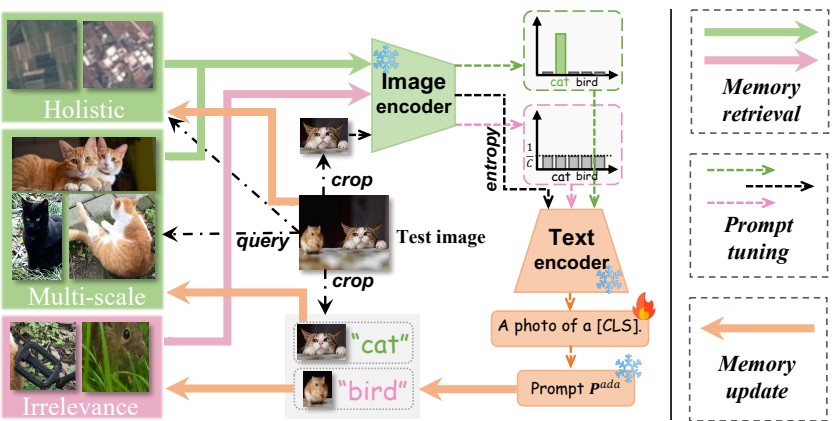

Figure 2: Overview of our method. Each test image undergoes three sequential steps: memory retrieval, prompt tuning, and memory update. In the memory retrieval step, the test image is used to query both the multi-scale memory and the holistic memory. The predicted label is then used to fetch class-relevant patches or images, as well as misleading patches from the class-irrelevant memory. During prompt tuning, the textual prompt is optimized using the test image and the retrieved visual memory. Finally, in the memory update step, the adapted prompt is used to update the class-relevant patches and the test image in the multi-scale and holistic memories, while high-confidence but irrelevant patches are added to the class-irrelevant memory.

classes and computes the probability of class membership as follows:

$$p(y = c \mid \mathbf{X}, \mathbf{p}) = \frac{\exp(\cos(f(\mathbf{X}), g(\mathbf{p}_c))/\tau)}{\sum_{j=1}^{C} \exp(\cos(f(\mathbf{X}), g(\mathbf{p}_j))/\tau)}, \tag{1}$$

where $\cos(\cdot, \cdot)$ denotes cosine similarity, and $\tau$ is a learned temperature parameter. For clarity, we assume that the outputs of the encoders are normalized by default throughout the rest of the paper.

**Test-time prompt tuning.** Directly applying CLIP to downstream tasks may suffer from performance degradation due to distribution shifts. Test-time prompt tuning (TPT) aims to adapt CLIP to the test data by optimizing a learnable prompt at test time. For instance, the pioneering TPT method [38] augments the test image $\mathbf{X}$ into $N$ views $\mathbf{X}_{[N]}$, and then optimizes a learnable prompt $\mathbf{p}$ using $n$ selected augmentations $\mathbf{X}_{[n]}$ with low entropy, based on an entropy minimization loss:

$$\mathcal{L} = -\sum_{c=1}^{C} \bar{p}(y = c \mid \mathbf{X}_{[n]}, \mathbf{p}) \cdot \log \bar{p}(y = c \mid \mathbf{X}_{[n]}, \mathbf{p}), \tag{2}$$

where $\bar{p}(y = c \mid \mathbf{X}_{[n]}, \mathbf{p})$ denotes the average prediction probability across the selected augmentations $\mathbf{X}_{[n]}$.

### 3.2 Multi-scale visual memory

Previous TPT methods typically use only the current test image or its augmentations to learn a tunable prompt. However, the visual class information available from a single test image is limited for prompt learning. As a result, TPT methods significantly underperform recent human-designed prompt methods, as shown in Fig. 1a. To address this limitation, we propose TPT enhanced by multi-scale visual memory, which provides diverse visual class information from past data to guide prompt learning. Specifically, our method integrates visual memory into the test-time prompt tuning workflow and introduces a **prompt-memory mutual promotion framework**. As illustrated in Fig. 2, for each test sample, the method involves three sequential steps: **Memory retrieval**, **Prompt tuning**, and **Memory update**.

**Memory retrieval.** Let the multi-scale visual memory be denoted as $\mathcal{M} \in \mathbb{R}^{C \times S \times D}$, where $C$ is the number of classes, $S$ is the memory size per class, and $D$ is the dimension of image patches. For

a given test image $\mathbf{X} \in \mathbb{R}^{3 \times H \times W}$, we compute the similarity between the memory patches and the image in the feature space encoded by the CLIP image encoder. Specifically, we denote the encoded test image as $f(\mathbf{X}) = \mathbf{v} \in \mathbb{R}^d$, and the encoded memory as $f(\mathcal{M}) = \mathbf{M} \in \mathbb{R}^{C \times S \times d}$. We define the $(c, m)$-th memory vector as $\mathbf{m}_{c,m} := \mathbf{M}[c, m] \in \mathbb{R}^d$. The similarity is computed as:

$$\mathbf{S}_{c,m} = \phi\left(\mathbf{v}^\top \mathbf{m}_{c,m}\right), \quad \text{for } c = 1, \ldots, C, \ m = 1, \ldots, S, \tag{3}$$

where $\phi$ is an exponential scaling function defined as $\phi(x) = \exp(-\beta(1 - x))$, as in [45]. We then identify the most similar class in the visual memory to the current test sample based on cosine similarity:

$$\tilde{y} = \arg\max_{c \in \{1, \ldots, C\}} \left(\mathbf{M}_c^{\text{ada}\top} \mathbf{v}\right), \quad \mathbf{M}_c^{\text{ada}} = \text{Norm}\left(\sum_{m=1}^{S} \mathbf{S}_{c,m} \cdot \mathbf{m}_{c,m}\right), \tag{4}$$

where the visual memory is weighted by $\mathbf{S}$ before the cosine similarity computation, following [49, 46], and Norm denotes $\ell_2$ normalization. Finally, we use the pseudo label $\tilde{y}$ to get the corresponding class-specific visual memory $\mathcal{M}_{\tilde{y}}$ as the retrieved class-relevant memory for the current test image.

**Prompt tuning.** In this step, we use the current test image $\mathbf{X}$ and the retrieved relevant visual memory $\mathcal{M}_{\hat{y}}$ to learn the textual prompt $\mathbf{p}_{\text{init}}$. For the test image, we apply an entropy minimization loss, as in TPT [38], shown in Eq. 2, where we adopt random cropping as the data augmentation strategy. Concurrently, we incorporate a cross-entropy loss between the retrieved memory and the pseudo label to enhance the prompt learning. Starting from $\mathbf{p}_{\text{init}}$, we optimize the prompt to obtain $\mathbf{p}_{\text{ada}}$:

$$\mathbf{p}_{\text{ada}} = \mathbf{p}_{\text{init}} - \eta \cdot \nabla_{\mathbf{p}} \mathcal{L}_{\text{pt}} = \mathbf{p}_{\text{init}} - \eta \cdot \nabla_{\mathbf{p}} \left[ \mathcal{H}\left(\bar{p}(\mathbf{X}_{[n]}, \mathbf{p})\right) - \log \bar{p}(y = \tilde{y} \mid \mathcal{M}_{\tilde{y}}, \mathbf{p}) \right] \tag{5}$$

where $\eta$ denotes the learning rate. $\mathbf{X}_{[n]}$ denotes $n$ cropped patches of $\mathbf{X}$ selected from the full set of $N$ patches $\mathbf{X}_{[N]}$ based on low prediction entropy. $\bar{p}(\cdot)$ denotes the average predicted probability over patches of the test image or memorized patches. $\mathcal{H}(\cdot)$ denotes the entropy of a predicted probability distribution $p(\cdot)$ over $C$ classes, defined as $\mathcal{H}(p) = -\sum_{c=1}^{C} p(y = c) \cdot \log p(y = c)$.

**Memory update.** This step aims to update the multi-scale visual memory $\mathcal{M}$ with the most relevant patch from the current test image, based on the adapted textual prompt $\mathbf{p}_{\text{ada}}$. Specifically, we select a patch from the $N$ randomly cropped views $\mathbf{X}_{[N]}$ according to vision-text similarity:

$$\hat{y} = \arg\max_{c} \bar{p}(y = c \mid \mathbf{X}_{[n]}, \mathbf{p}_{\text{ada}}), \quad i^* = \arg\min_{i \in \mathcal{I}} \mathcal{H}(p(\{\mathbf{X}_{[N]}\}_i, \mathbf{p}_{\text{ada}})), \tag{6}$$

$$\text{where} \quad \mathcal{I} = \left\{ j : \arg\max_{c} p(y = c \mid \{\mathbf{X}_{[N]}\}_j, \mathbf{p}_{\text{ada}}) = \hat{y} \right\}. \tag{7}$$

We first obtain a confident prediction $\hat{y}$ by aggregating predictions over the selected subset $\mathbf{X}_{[n]}$ using the adapted prompt $\mathbf{p}_{\text{ada}}$. Then, from the subset $\mathcal{I}$ of patches whose predicted label matches $\hat{y}$, we select the patch $\mathbf{X}_{i^*}$ with the lowest prediction entropy. This avoids directly selecting the lowest-entropy patch from the entire set $\mathbf{X}_{[N]}$, which may include highly confident but irrelevant patches. Finally, we insert the selected patch into the corresponding memory slot $\mathcal{M}_{\hat{y}}$. If the memory is at full capacity, we remove the patch with the highest entropy among the existing entries and the current candidate.

These three steps for each test image constitute a round of mutual promotion between the tunable textual prompt and the evolving visual memory. Afterward, we obtain two predictions for the current test image: one from the optimized prompt and one from the updated memory $\mathcal{M}'$. We combine them to produce the final prediction:

$$P_{\text{final}} = P_{\text{pt}} + P_{\text{memo}} = p(\mathbf{y} \mid \mathbf{v}, \mathbf{p}_{\text{ada}}) + \text{Softmax}(\mathbf{M}'^{\text{ada}\top} \mathbf{v}), \tag{8}$$

where $P_{\text{pt}}, P_{\text{memo}} \in \mathbb{R}^C$. The prediction $P_{\text{memo}}$ is obtained via similarity-based classification, as in the memory retrieval step, and $\mathbf{M}'^{\text{ada}}$ is computed from the updated memory following Eqs. 3 and 4.

It is worth noting that we perform only a single forward pass of the CLIP image encoder for each test image and its patches, as the image encoder is frozen during the test-time prompt tuning process. The encoded visual features are reused across all three steps, such as in Eqs. 4, 5, and 6. Therefore, we directly store the encoded features in the multi-scale visual memory, i.e., $\mathbf{M} \in \mathbb{R}^{C \times S \times d}$, in practice.

## 3.3 Holistic visual memory

In downstream tasks, there are not only object recognition tasks but also holistic visual recognition tasks, such as land cover classification [41] and scene understanding [13]. These tasks require holistic, image-level information, which may be lost when using only image patches. Accordingly, we introduce a holistic visual memory that works in coordination with the aforementioned multi-scale visual memory.

During memory retrieval, we use the current test image as a query to retrieve relevant visual memory from both the multi-scale memory and the holistic memory, i.e., $\{\mathcal{M}, \mathcal{M}^{\mathrm{hol}}\}$. Specifically, we compute the similarity-based probability distribution $\mathrm{Softmax}(\mathbf{M}^{\mathrm{ada}\top}\mathbf{v})$ using both types of memory and select the one with lower entropy to fetch the class-relevant visual memory. The prompt tuning step remains unchanged, except that the retrieved memory used in Eq. 5 is selected from either the multi-scale or holistic memory. During memory update, both types of memory update the same memory slot, $\mathcal{M}_{\hat{y}}$ and $\mathcal{M}^{\mathrm{hol}}_{\hat{y}}$, as determined by the mechanisms in Eqs. 6 and 7. In addition, the holistic visual memory also contributes to the memory-based prediction $P_{\mathrm{memo}}$, producing a prediction in the same way as the multi-scale memory. We then select the one with lower entropy as the final $P_{\mathrm{memo}}$ in Eq. 8.

## 3.4 Irrelevance suppression

The memory retrieval and memory update processes operate without ground truth supervision at test time, making the memory inevitably noisy. To mitigate the adverse impact, we design an irrelevance suppression strategy: selectively retrieving and using class-relevant memory, while proactively penalizing class-irrelevant memory. Specifically, during memory retrieval, we filter out relatively irrelevant memory based on the similarity matrix $\mathbf{S}$:

$$\mathbf{M}^{\mathrm{top}}_{\tilde{y}} = \mathbf{M}_{\tilde{y}}\left[\mathrm{TopK}\left(\mathbf{S}_{\tilde{y}}, \lfloor|\mathbf{M}_{\tilde{y}}|\cdot\gamma\rfloor\right)\right], \tag{9}$$

where $\mathrm{TopK}(\cdot, k)$ returns the indices of the top $k$ elements with the highest similarity scores. $|\mathbf{M}_{\tilde{y}}|$ denotes the number of stored features in memory for class $\tilde{y}$, and $\gamma \in (0, 1]$ is the selection ratio. The filtered memory $\mathbf{M}^{\mathrm{top}}_{\tilde{y}}$ is then used in the prompt tuning stage (see Eq. 5).

In addition, we construct a class-irrelevant memory $\mathcal{M}^{\mathrm{irr}}$ to store previously seen, misleading visual cues from the test domain. Technically, we update this memory with high-confidence patches that are estimated to be irrelevant. Specifically, after the multi-scale memory update, given the memory prediction for the test image $\hat{y}_{\mathrm{memo}} = \arg\max_c P_{\mathrm{memo}}$, and the optimized-prompt-based predictions of patches $\hat{\mathbf{y}}^N = \arg\max_c p(y = c \mid \mathbf{X}_{[N]}, \mathbf{p}_{\mathrm{ada}})$, $\quad \hat{\mathbf{y}}^n = \arg\max_c p(y = c \mid \mathbf{X}_{[n]}, \mathbf{p}_{\mathrm{ada}})$, if the memory prediction and the predictions of selected patches are consistent, i.e., $\sum_{j=1}^n \mathbb{1}\left[\hat{\mathbf{y}}^n_j = \hat{y}_{\mathrm{memo}}\right] = n$, we regard $\hat{y}_{\mathrm{memo}}$ as a confident prediction. Then, the irrelevant memory is updated as:

$$i^* = \arg\min_{i\in\mathcal{I}} \mathcal{H}(p(\{\mathbf{X}_{[N]}\}_i, \mathbf{p}_{\mathrm{ada}})), \quad \text{where} \quad \mathcal{I} = \left\{i : \hat{\mathbf{y}}^N_i \neq \hat{y}_{\mathrm{memo}}\right\}. \tag{10}$$

Here, we select the highest-confidence patch $\{\mathbf{X}_{[N]}\}_{i^*}$ among those that disagree with the confident prediction and store it in the memory slot $\mathbf{M}^{\mathrm{irr}}_{\hat{\mathbf{y}}^N_{i^*}}$.

The class-irrelevant memory stores patches with pseudo "wrong" labels—i.e., patches that are confidently predicted to belong to a different class. This contradicts the task assumption. For example, in a label space of "cat" and "bird", an image labeled "cat" is not expected to contain a bird. To suppress these confident but irrelevant cues, we apply a flat-label KL loss:

$$\mathcal{L}_{\mathrm{irr}} = \min\left(\frac{\alpha}{C}, \beta\right)\mathrm{KL}\left(\frac{1}{C}\mathbf{1} \,\middle\|\, p(y = \tilde{y} \mid \mathcal{M}^{\mathrm{irr}}_{\tilde{y}}, \mathbf{p})\right), \tag{11}$$

where $\alpha$ and $\beta$ are hyperparameters, and $C$ is the number of classes. This loss $\mathcal{L}_{\mathrm{irr}}$ is incorporated into the prompt tuning objective $\mathcal{L}_{\mathrm{pt}}$.

# 4 Experiments

## 4.1 Experimental setup

**Datasets.** Following prior test-time prompt tuning methods [38, 42], we evaluate our method on 15 datasets, including downstream image classification tasks and out-of-distribution benchmark datasets.

Table 1: **Results on 10 downstream image classification datasets.** The reported numbers are top-1 accuracy (%). Methods marked with * include training with labeled data from ImageNet. M$^2$TPT$^\dagger$ represents the version that incorporates hand-crafted and LLM-generated prompts.

| Method | Venue | Flower | DTD | Pets | Cars | UCF | Caltech | Food | SUN | Aircraft | EuroSAT | Average |
|---|---|---|---|---|---|---|---|---|---|---|---|---|
| CLIP [35] | - | 67.44 | 44.27 | 88.25 | 65.48 | 65.13 | 93.35 | 83.65 | 62.59 | 23.37 | 42.01 | 63.55 |
| *Prompt-Tuning-Based Methods* | | | | | | | | | | | | |
| CoOp * [48] | IJCV22 | 68.71 | 41.92 | 89.14 | 64.51 | 66.55 | 93.70 | 85.30 | 64.15 | 18.47 | 46.39 | 63.88 |
| CoCoOp * [47] | CVPR22 | 71.88 | 45.73 | 90.14 | 65.32 | 68.21 | 94.43 | 86.06 | 67.36 | 22.94 | 45.37 | 65.74 |
| MaPLe * [19] | CVPR23 | 72.23 | 46.49 | **90.49** | 65.57 | 68.69 | 93.53 | 86.20 | 67.01 | 24.74 | 48.06 | 66.20 |
| TPT [38] | NeurIPS22 | 68.98 | 47.75 | 87.79 | 66.87 | 68.04 | 94.16 | 84.67 | 65.50 | 24.78 | 42.44 | 65.20 |
| DiffTPT [10] | ICCV23 | 70.10 | 47.00 | 88.20 | 67.01 | 68.22 | 92.49 | **87.23** | 65.74 | **25.60** | 43.13 | 65.47 |
| C-TPT [43] | ICLR24 | 69.80 | 46.00 | 88.20 | 65.80 | 65.70 | 93.60 | 83.70 | 64.80 | 24.00 | 43.20 | 64.80 |
| DynaPrompt [42] | ICLR25 | 69.95 | 47.96 | 88.28 | 67.65 | 68.72 | **94.32** | 85.42 | 66.32 | 24.33 | 42.28 | 65.52 |
| **M$^2$TPT** | - | **73.65** | **50.24** | 89.48 | **68.91** | **71.42** | 93.35 | 86.63 | **68.12** | 23.46 | **59.14** | **68.44** |
| *Methods Using Hand-Crafted and LLM-Generated Prompts* | | | | | | | | | | | | |
| VisDesc [29] | ICLR23 | 70.85 | 44.98 | 88.85 | 64.08 | 67.12 | 94.60 | 85.05 | 67.99 | 24.30 | 54.84 | 66.27 |
| WaffleCLIP [37] | ICCV23 | 72.35 | 45.21 | 89.95 | 63.57 | 67.19 | 94.02 | 86.68 | 67.23 | 25.39 | 55.07 | 66.67 |
| CuPL [34] | ICCV23 | 71.30 | 44.56 | 89.13 | 65.29 | 66.83 | 92.98 | 86.11 | 62.59 | 24.90 | 47.84 | 65.15 |
| TDA [18] | CVPR24 | 71.42 | 47.40 | 88.63 | 67.28 | 70.66 | 94.24 | 86.14 | 67.62 | 23.91 | 58.00 | 67.53 |
| DMN [46] | CVPR24 | 74.49 | **55.85** | 92.04 | 67.96 | 72.51 | 95.38 | 86.00 | 70.18 | 30.03 | 59.43 | 70.40 |
| AWT [50] | NeurIPS24 | 75.07 | 55.56 | 92.53 | 69.93 | 72.51 | **95.54** | 85.54 | 70.58 | 29.22 | 58.61 | 70.51 |
| **M$^2$TPT$^\dagger$** | - | **76.90** | 55.32 | **92.31** | 69.32 | **74.25** | 94.24 | **86.42** | 70.65 | 30.48 | 62.32 | **71.34** |

Table 2: **Results on out-of-distribution benchmark datasets.** The marked M$^2$TPT$^\dagger$ represents the version that incorporates hand-crafted and LLM-generated prompts.

| Method | Venue | ImageNet | ImageNet-A | ImageNet-V2 | ImageNet-R | ImageNet-S | OOD Average | Average |
|---|---|---|---|---|---|---|---|---|
| CLIP [35] | - | 66.73 | 47.87 | 60.86 | 73.98 | 46.09 | 57.20 | 59.11 |
| *Prompt-Tuning-Based Methods* | | | | | | | | |
| TPT [38] | NeurIPS22 | 68.98 | 54.77 | 63.45 | 77.06 | 47.94 | 60.80 | 62.44 |
| DiffTPT [10] | ICCV23 | 70.30 | 55.68 | **65.10** | 75.00 | 46.80 | 60.64 | 62.58 |
| C-TPT [43] | ICLR24 | 69.30 | 52.90 | 63.40 | 78.00 | 48.50 | 60.70 | 62.42 |
| DynaPrompt [42] | ICLR25 | 69.61 | 56.17 | 64.67 | **78.17** | 48.22 | 61.81 | 63.37 |
| **M$^2$TPT** | - | **71.49** | **60.11** | 64.82 | 76.79 | **50.79** | **63.13** | **64.80** |
| *Methods Using Hand-Crafted and LLM-Generated Prompts* | | | | | | | | |
| VisDesc [29] | ICLR23 | 68.55 | 49.07 | 61.80 | 75.13 | 47.97 | 58.49 | 60.50 |
| WaffleCLIP [37] | ICCV23 | 68.81 | 50.78 | 62.54 | 77.49 | 49.10 | 59.98 | 61.74 |
| CuPL [34] | ICCV23 | - | 50.72 | 63.27 | 77.05 | 49.02 | 60.02 | - |
| TDA [18] | CVPR24 | 69.51 | 60.11 | 64.67 | 80.24 | 50.54 | 63.89 | 65.01 |
| DMN [46] | CVPR24 | 72.25 | 58.28 | 65.17 | 78.55 | **53.20** | 63.80 | 65.49 |
| AWT [50] | NeurIPS24 | 71.32 | 60.33 | 65.15 | **80.64** | 51.60 | 64.43 | 65.81 |
| **M$^2$TPT$^\dagger$** | - | **73.01** | **62.55** | **65.86** | 77.48 | 53.03 | **64.73** | **66.39** |

The downstream image classification datasets include Flowers102 [31], DTD [6], OxfordPets [33], StanfordCars [21], UCF101 [39], Caltech101 [9], Food101 [4], SUN397 [41], FGVC-Aircraft [28], and EuroSAT [13]. For the out-of-distribution benchmark, we include ImageNet [7] and its four variants exhibiting domain shifts: ImageNet-A [15], ImageNet-V2 [36], ImageNet-R [14], and ImageNet-S [40]. For ImageNet, we use the validation set for evaluation, and adopt the same dataset splits as in TPT [38] for the remaining 14 datasets.

**Implementation details.** We use CLIP [35] with the ViT-B/16 encoder [8] for all experiments. For each test image, our method optimizes the textual prompt with a single update step, starting from the generic prompt "a photo of a [CLASS]." We use the AdamW optimizer [26] with a learning rate of $\eta = 0.003$ across all datasets. For random cropping, the scale range and aspect ratio range are set to $(0.08, 1)$ and $\left(\frac{3}{4}, \frac{4}{3}\right)$, respectively. The number of random crops $N$ is set to 32 for downstream classification datasets and 64 for out-of-distribution benchmark datasets, with a selection ratio of $n/N = 0.1$. For all datasets, the memory size $S$ is set to 50, and the hyperparameters for irrelevance suppression are set to $\gamma = 0.5$, $\alpha = 5$, and $\beta = 0.1$.

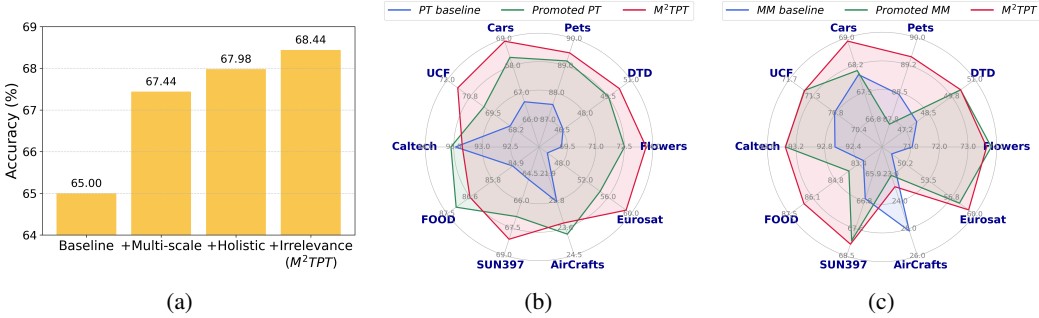

(a)                          (b)                          (c)

Figure 3: **(a) Ablation on main components.** The baseline is test-time prompt tuning using only low-entropy image patches. We incrementally add the three main components of our method and illustrate the performance gain contributed by each. **(b), (c) Analysis of the mutual promotion between the learnable prompt and the evolving memory.** In (b), we compare the standard test-time prompt tuning baseline with predictions $P_{\text{pt}}$ obtained from the prompt learned with visual memory. In (c), the memory-based predictions $P_{\text{memo}}$ are compared with a baseline where memory is updated using a static prompt, "a photo of a [CLS]." Improvements over the baselines demonstrate the mutual promotion between the learnable prompt and the visual memory.

## 4.2   Comparisons

We compare our test-time prompt tuning (TPT) method with recent TPT approaches, including TPT [38], DiffTPT [10], C-TPT [43], and DynaPrompt [42], and prompt learning methods including CoOp [48], CoCoOp [47], and MaPLe [19]. These methods, like ours, do not involve hand-crafted or LLM-generated prompts. Moreover, we also compare our method with recent VLM adaptation approaches that utilize hand-crafted or LLM-generated prompts, including VisDesc [29], Waffle-CLIP [37], CuPL [34], TDA [18], DMN [46], and AWT [50]. For this comparison, we design a variant of our method by simply incorporating human-designed prompts used in DMN [46] into the memory update step. Specifically, the confident prediction $\hat{y}$ in Eq. 6 is obtained by combining predictions from the adapted prompt and the human-designed prompts.

**Comparisons on downstream classification tasks.**   Tab. 1 presents results on 10 downstream fine-grained classification datasets. The upper part of the table compares prompt-tuning-based methods that learn a trainable prompt from a generic initialization. Compared to previous TPT methods, our method achieves the highest accuracy on 7 datasets and yields an average improvement of $2.92\%$. Notably, M$^2$TPT outperforms previous TPT methods by $15.94\%$ on the EuroSAT dataset. The lower part of the table compares methods that utilize hand-crafted and LLM-generated prompts. First, we observe that our method can benefit from incorporating human-designed prompts, achieving an average improvement of $2.9\%$. Compared to these methods, M$^2$TPT performs best on 8 out of 10 datasets and surpasses the second-best method by an average margin of $0.83\%$.

**Comparisons on out-of-distribution datasets.**   Tab. 2 shows results on ImageNet and four out-of-distribution datasets that exhibit distribution shifts from ImageNet. In the upper part of the table, M$^2$TPT outperforms recent test-time prompt tuning methods with an average improvement of $1.43\%$ across the five datasets. As shown in the lower part, M$^2$TPT also achieves state-of-the-art performance among VLM adaptation methods that leverage hand-crafted and LLM-generated prompts.

## 4.3   Ablation studies

**Ablation on main components.**   We study the effectiveness of the three components in M$^2$TPT—multi-scale visual memory, holistic visual memory, and irrelevance suppression—introduced in Secs. 3.2, 3.3, and 3.4, respectively, across 10 downstream classification datasets. We begin with a baseline that performs prompt tuning alone using selected low-entropy patches, as shown in Fig. 3a. Adding multi-scale visual memory to the baseline establishes the core framework of our method and improves the average accuracy to $67.44\%$, yielding a $2.44\%$ gain. Next, we incorporate holistic visual memory, which preserves global visual context for tasks that require holistic visual understanding, resulting in a further $0.54\%$ improvement. Finally, we introduce the

irrelevance suppression strategy to better exploit the noisy test-time memory, increasing the accuracy from $67.98\%$ to $68.44\%$.

**Analysis of the mutual promotion between the learnable prompt and the evolving memory.** In $M^2$TPT, the visual memory provides class-relevant visual descriptions to enhance textual prompt learning, and reciprocally, the learned prompt helps update the visual memory. To verify this mutual promotion effect, we design two baselines: the prompt tuning (PT) baseline and the memory (MM) baseline. The PT baseline corresponds to standard prompt tuning with selected low-entropy image patches. Its performance on 10 downstream classification datasets is shown in Fig. 3b. In the figure, Promoted PT refers to the performance of the learned prompt enhanced by visual memory, corresponding to $P_{\mathrm{pt}}$ in Eq. 8. Compared to the PT baseline, Promoted PT consistently demonstrates superior performance, highlighting the improvement in prompt tuning enabled by the multi-scale visual memory. In Fig. 3c, the MM baseline denotes a memory-based method where the memory is updated using a generic prompt "a photo of a [CLASS]." In contrast, Promoted MM refers to the prediction generated from the evolving memory updated with the learned prompt, i.e., $P_{\mathrm{memo}}$ in Eq. 8. Promoted MM outperforms the MM baseline on 8 out of 10 datasets, indicating the beneficial effect of the learned prompt on memory updates. Finally, $M^2$TPT achieves consistently better performance than both Promoted PT and Promoted MM, demonstrating the effectiveness of combining the two predictions as defined in Eq. 8.

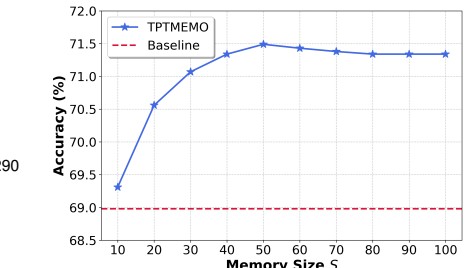

Figure 4: **Effect of memory size.**

Table 3: **Computation resources of test-time prompt tuning methods.** The methods are evaluated on the DTD dataset using an RTX A4500 GPU.

| Method | Memory (GB) | Runtime (s) |
|---|---|---|
| TPT | 1.56 | 0.14 |
| TPT (bs=32) | 1.53 | 0.11 |
| DynaPrompt | 9.98 | 0.41 |
| $M^2$TPT | 1.66 | 0.11 |

**Effect of memory size.** We study the effect of memory size $S$ on the validation set of the ImageNet dataset. As shown in Fig. 4, the Baseline refers to test-time prompt tuning without memory. $M^2$TPT shows increasing accuracy as $S$ increases from 10 to 50, consistently outperforming the Baseline. When the memory size exceeds 50, the accuracy saturates and slightly decreases.

**Computation resource.** We compare GPU memory usage and runtime of $M^2$TPT with recent test-time prompt tuning methods, as shown in Tab. 3. All results are measured on the DTD dataset using an RTX A4500 GPU. DynaPrompt [42] introduces significantly higher memory usage and longer runtime than other methods because it optimizes multiple prompts. Compared to TPT [38] under its official setting (with augmentation batch size 64), $M^2$TPT uses only 0.1 GB more GPU memory while consuming less runtime per image. When we test TPT with the same batch size (bs=32) as $M^2$TPT, the runtime becomes comparable, and the memory usage difference increases to 0.13 GB. These results suggest that the visual memory module in $M^2$TPT introduces only a small memory overhead and minimal impact on runtime.

## 5  Conclusion

In this paper, we identified a core limitation of previous TPT methods: learning prompts from limited visual information provided by the current test image, which makes the learned prompts less competitive compared to prompt-engineering-based approaches. To address this, we proposed test-time prompt tuning with multi-scale visual memory, enabling the model to learn prompts from both class-relevant visual descriptions observed in the past and the current test image. Extensive experiments demonstrate that our method outperforms existing TPT methods while introducing minimal additional computational cost. Moreover, our method can benefit from prompt engineering and achieves state-of-the-art performance compared to recent prompt-engineering-based VLM adaptation methods by incorporating human-designed prompts into our framework.

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
