# OpenReview forum: "Echoes of the Visual Past: Test-Time Prompt Tuning with Multi-Scale Visual Memory"
_NeurIPS.cc/2025/Conference — Submitted to NeurIPS 2025_

### Official Review · Reviewer_e7b1 · 2025-06-20

**Clarity:** 2
**Significance:** 3
**Originality:** 2
**Rating:** 4
**Confidence:** 3

**Summary:**

This paper proposes a new test-time prompt tuning method for CLIP, termed M^2TPT. The proposed method uses additional visual information provided by multi-scale memory to enhance the learning effect, and uses an irrelevance suppression strategy is designed to eliminate potential noise data in the memory.

**Questions:**

1) How is multi-scale visual memory initialized?
2) There is a lack of explanation for the methods between lines 145 and 151. The authors should explain how these methods from [45, 46, 49] contribute to their work.
3) The author should strengthen the checking of symbolic definitions. For example, the size of the matrix P_{ada} is not given; L_{pt} in Formula 5 is not defined.
4) The author needs to confirm the correctness of the formula. In Eq.8, p(y|v,p_{ada}) represents the probability of belonging to y, which should be a value rather than a distribution.
5) The author believes that the bottleneck of existing methods is that they only use visual information from the current image, and introduces memory to solve this problem. To fully illustrate this problem, the author's baseline should add a multi-scale patch from the current image to illustrate the gain brought by memory through historical information.

**Ethical Concerns:**

["NO or VERY MINOR ethics concerns only"]

**Final Justification:**

The authors' supplementary notes explain my doubts about the source of the performance and clarify any unclear points in the paper.

**Limitations:**

yes

**Quality:**

2

**Strengths And Weaknesses:**

Strengths:
1) The proposed M^2TPT method achieves great performance improvement.

Weakness:
1) Thesis writing needs to be improved, see question 2-4.
2) The motivation for the work is not fully verified, see question 5.

---

> ### Author Rebuttal · Authors · 2025-07-31
>
> We thank the reviewer for the constructive comments and suggestions. We appreciate the reviewer’s recognition of the great performance improvements achieved by our work. Below, we respond to all weaknesses (W) and questions (Q) point by point.
>
> > Q1: How is multi-scale visual memory initialized?
>
> The multi-scale visual memory is empty at the beginning of the test phase. It accumulates visual memory progressively as test data streams in. Once the memory reaches its capacity, the patch with the lowest confidence is removed to maintain the fixed memory size.
>
> > Q2 & W1: There is a lack of explanation for the methods between lines 145 and 151. The authors should explain how these methods from [45, 46, 49] contribute to their work.
>
> For [45], we adopt its proposed exponential scaling function to convert the computed similarity in Eq. (3) into non-negative values and to modulate the sharpness of the similarity scores. For [46] and [49], we follow their approach to take into account the relationship between the memory and the test image when using memory for prediction. We will further clarify these connections in the revised manuscript.
>
> > Q3 & W1: The author should strengthen the checking of symbolic definitions. For example, the size of the matrix P_{ada} is not given; L_{pt} in Formula 5 is not defined.
>
> p_{ada} denotes the optimized textual prompt. Its size depends on the embedding size of the CLIP backbone used, and remains constant throughout the process. While not explicitly specified, its size does not affect other mathematical formulations in the framework. As for L_{pt}, it refers to the prompt optimization loss, which is defined on the right-hand side of Eq. (5).
>
> > Q4 & W1: The author needs to confirm the correctness of the formula. In Eq.8, p(y|v,p_{ada}) represents the probability of belonging to y, which should be a value rather than a distribution.
>
> Thank you for pointing out the typo. The expression in Eq. (8) should be $P$($\mathbf{y}$|$\mathbf{v}, \mathbf{p}_{ada}$)   ($P$ instead of $p$). It denotes the predicted distribution over classes with the adapted prompt.
>
>
> > Q5 & W2: The author believes that the bottleneck of existing methods is that they only use visual information from the current image, and introduces memory to solve this problem. To fully illustrate this problem, the author's baseline should add a multi-scale patch from the current image to illustrate the gain brought by memory through historical information.
>
> We have added such a baseline, named Multi-scale Patch, in the table below. This baseline uses multi-scale patches from the current test image for prompt tuning. The comparison demonstrates that our framework achieves notable performance improvements, confirming the effectiveness of the historical visual memory.
>
> | Method | Flower | DTD | Pets | Cars | UCF | Caltech | Food | SUN | Aircraft | EuroSAT | Average |
> |--------|--------|-----|------|------|-----|---------|------|-----|----------|---------|---------|
> |  Multi-scale Patch |69.18|46.57|87.57|66.67|68.57|93.47|85.00|64.80|22.77|45.44|65.00|
> |**M$^2$TPT**|**73.65**|**50.24**|**89.48**|**68.91**|**71.42**|**93.35**|**86.63**|**68.12**|**23.46**|**59.14**|**68.44**|

---

> ### Author Response · Authors · 2025-08-05
> **Thank You and Gentle Reminder**
>
> Dear Reviewer e7b1,
>
> We sincerely appreciate your valuable feedback and the time you have taken to review our work. As the discussion period will end within 48 hours, we just wanted to kindly check if you might have any further questions or points you would like to discuss.
>
> We would be very happy to provide additional clarification or details should you need them.
>
> Best regards,
> The Authors

---

> ### Author Response · Authors · 2025-08-07
> **Thank You for Your Review**
>
> Dear Reviewer e7b1,
>
> Thank you again for your valuable review of our work. We understand the concerns raised in your initial review, primarily including the need for an ablation study, further explanation of some related works, and a formula typo. We have addressed each of these points in our rebuttal by adding the requested ablation study and providing detailed clarifications.
>
> We just wanted to follow up to see if you have any further questions or comments. We would be happy to discuss them here before the end of the discussion phase.
>
> Best regards,\
> The Authors

---

> ### Comment · Area_Chair_gkqa · 2025-08-08
>
> Dear Reviewer e7b1,
>
> This is the last call for author-reviewer discussion, which will ends on Aug 8. Could you please read authors' rebuttal and confirm whether your concerns have been addressed asap? Thank you.
>
> Best,
>
> AC

---

> > ### Comment · Reviewer_e7b1 · 2025-08-08
> >
> > I apologize for my lateness.
> >
> > The author has addressed most of my concerns. However, I'm unsure of the article's contribution to the field, so I've remained silent.
> >
> > Based on the author's discussions with other reviewers, the article is insightful and informative, so I'm inclined to raise the score to 4. However, the author needs to further polish the article.

---

> > > ### Author Response · Authors · 2025-08-08
> > >
> > > Dear Reviewer e7b1,
> > >
> > > Thank you for your response. We are glad to hear that our rebuttal addressed your concerns, and we sincerely appreciate your decision to raise your rating to a positive score. We will further polish the paper in the revision, and should you have any other questions, we would be happy to discuss them.
> > >
> > > Best regards,
> > >
> > > The Authors

---

### Official Review · Reviewer_rbTk · 2025-06-29

**Clarity:** 3
**Significance:** 3
**Originality:** 3
**Rating:** 4
**Confidence:** 4

**Summary:**

This paper proposes Test-time Prompt Tuning with Multi-scale visual Memory (M2TPT). Specifically, the memory is constructed to store past seen class-relevant image patches as multi-scale visual descriptions for each class. For each test image, they use it to query the memory and learn the textual prompt using both the test image and the retrieved class-relevant visual memory. Additionally, they introduce holistic visual memory to better handle holistic visual recognition tasks that require global image-level context, and an irrelevance suppression strategy to mitigate the impact of noisy memory entries at test time. Experiments demonstrates its effectiveness.

**Questions:**

1. The authors appear to store multiple memory patches per class to facilitate memory retrieval. However, when the number of classes becomes large, such as up to 1000 classes, this design could result in significant memory overhead. How scalable is the proposed memory architecture in such high-class-count scenarios?

2. Moreover, since this memory retrieval process is executed at test time, would a large number of classes slow down the retrieval process and reduce retrieval accuracy? The scalability of both retrieval speed and precision under large-class settings needs to be clarified.

3. In the test-time prompt tuning stage, the method seems to assume that each input image belongs to one of the predefined C classes. However, CLIP is known for its strong generalization capabilities beyond closed-label settings. What happens if the input image does not belong to any of the predefined categories? How would this affect memory retrieval, prompt tuning, and final predictions? This limitation should be discussed, especially in the context of real-world open-set scenarios.

**Ethical Concerns:**

["NO or VERY MINOR ethics concerns only"]

**Final Justification:**

I believe this paper has its merits. However, after carefully reading the discussion between Reviewer D3JX and the authors, I think the paper’s comparison aspect should be expanded to make it more complete. Therefore, I am maintaining my borderline accept rating, but I will highlight the key points raised by Reviewer D3JX, which the authors should address and improve in the camera-ready version.

**Limitations:**

Please refers to Weakness and questions sections.

**Quality:**

3

**Strengths And Weaknesses:**

Strength:

1. The paper is well-written and easy to follow.

2. The ablation study is detailed and enough, can demonstrate the effectiveness of each component.

3. The discussed topic is deserved to explore and the proposed approach is novel.

Weakness:

1. The proposed method performs instance-level optimization during test-time inference. However, since the memory module is continually updated throughout the testing phase, early test samples and later ones rely on significantly different memory states. This raises a concern about performance imbalance across the test set: could the final results be sensitive to the order of test-time inputs? If so, this would compromise the robustness and reproducibility of the method, as test-time performance might fluctuate depending on input ordering.

2. In the irrelevance suppression stage, high-confidence but incorrectly predicted samples are possible to be added to the class-irrelevant memory. If such mispredictions occur early in the test phase, they may lead to cascading memory contamination, where future retrieval is misled by accumulated errors, reinforcing incorrect representations. How does the method mitigate this risk?

3. The framework appears to lack a long-term memory management strategy. As tuning progresses, the memory continues to grow, but the paper does not clarify whether there is a limit on memory size, a mechanism for discarding outdated samples, or any decay strategy for less relevant entries. Without such mechanisms, there is a risk that the memory will become bloated or conflicted, ultimately degrading its quality and representativeness. How is this issue addressed in the proposed design?

---

> ### Author Rebuttal · Authors · 2025-07-31
>
> We thank the reviewer for the constructive comments and suggestions. We appreciate the reviewer’s recognition of our work for its novelty, thorough ablation studies, and clear writing. Below, we respond to all weaknesses (W) and questions (Q) point by point.
>
> > W1: Could the final results be sensitive to the order of test-time inputs? If so, this would compromise the robustness and reproducibility of the method, as test-time performance might fluctuate depending on input ordering.
>
> We have included an analysis of different input test data orders in Appendix B, and we present the results in the table below. Specifically, we run three experiments with different random seeds that change the input test data order. The results show acceptable variance. We will also add a note in the main body of the paper.
> That said, we acknowledge a relevant limitation, discussed in Appendix D: memory-based methods initialize the memory with early input data, so a high volume of low-quality or difficult early samples may lead to suboptimal memory updates and introduce significant noise into the memory.
>
> ||$P_{pt}$|$P_{memo}$|$P_{final}$|
> |---|---|---|---|
> |Mean (std)|67.46 (0.07)|67.42 (0.05)| 68.24 (0.14)|
>
>
> > W2: If mispredictions occur early in the test phase, they may lead to cascading memory contamination, where future retrieval is misled by accumulated errors, reinforcing incorrect representations. How does the method mitigate this risk?
>
> Memory-based methods indeed face the risk of memory contamination due to mispredictions during test-time updates. Unlike previous methods that update the memory using fixed CLIP representations, our method performs memory updates after adapting CLIP via prompt tuning to the current test data. This reduces the risk of early-stage mispredictions overwhelming the memory and leads to more robust memory updates. Experimentally, our proposed method is superior compared to DMN[46] and TDA [18] where fixed CLIP representations are utilized to construct the memory.
>
>
> > W3: The framework appears to lack a long-term memory management strategy, specifically a limit on memory size and a mechanism for discarding outdated samples. Without such mechanisms, there is a risk that the memory will become bloated or conflicted.
>
> We would like to clarify that our method does include a memory management mechanism with a fixed memory size of 50 entries per class, as described in lines 168–170 of the Method section and in the Implementation Details. During the memory update step, if a class’s memory has not reached its capacity, the current sample is stored directly. If the memory is full, we remove the sample with the lowest prediction confidence (highest entropy) among the current and existing entries.
>
> > Q1 & Q2: How do additional memory overhead and runtime introduced by the memory part (including memory retrieval) increase with class number increasing? Would a large number of classes lead to retrieval accuracy degradation?
>
> We analyze the additional memory overhead and runtime introduced by the memory component (including retrieval) on both low- and high-class-count datasets, shown in the table below. As expected, the overhead increases with the number of classes. However, it remains a small fraction of the overall system cost compared to the prompt tuning part, which adds ~15.8 GB when scaling from 47 to 1000 classes.
>
> ||Additional Memory (GB) |Additional Runtime (s)|
> |---|---|---|
> |DTD (47 classes)|0.13|0.01|
> |ImageNet (1000 classes)|1.44|0.09|
>
> Memory retrieval accuracy influences the contribution of the memory part to the overall system. To assess whether memory retrieval accuracy degrades as the number of classes increases, we report the performance gains from the memory part across datasets with varying class counts. The table below shows consistent improvements, suggesting that memory retrieval remains effective and scalable even with 1000 classes.
>
> | |UCF101 (101 classes)|SUN397(397 classes)| ImageNet(1000 classes)|
> |-----|-----|-----|-----|
> |w/o memory| 68.57| 64.80| 69.04|
> |w memory| 71.42 (+ 2.85) | 68.12 (+ 3.32)| 71.49 (+ 2.45)|
>
>
> > Q3: The method seems to assume a closed-set scenario. This limitation should be discussed, especially in the context of real-world open-set scenarios.
>
> We have noted this limitation in the Limitations section: this work addresses the closed-set setting, ***in line with most previous test-time prompt tuning methods.*** The current framework does not consider open-set test data streams. As future work, one promising direction is to incorporate recent out-of-distribution (OOD) detection techniques, which can identify and filter out samples not belonging to the predefined classes [1].
>
> [1] Generalized Out-of-Distribution Detection and Beyond in Vision Language Model Era: A Survey. In Transactions on Machine Learning Research. 2025.

---

> > ### Comment · Reviewer_rbTk · 2025-08-05
> >
> > I have read the authors rebuttal, and my concerns are mostly addressed, I would maintain my original score.

---

> ### Author Response · Authors · 2025-08-07
> **Thank you for your response**
>
> Dear Reviewer rbTk,
>
> Thank you for your response and for your thoughtful feedback throughout the review process. We're glad to hear that our rebuttal addressed your concerns and deeply appreciate that you remain positive on our submission. If you have any further questions or thoughts, we’d be happy to discuss them before the end of the discussion phase.
>
> Best regards,\
> The Authors

---

### Official Review · Reviewer_D3JX · 2025-06-29

**Clarity:** 2
**Significance:** 2
**Originality:** 2
**Rating:** 3
**Confidence:** 4

**Summary:**

This paper introduces "Test-Time Prompt Tuning with Multi-Scale Visual Memory" (M2TPT in short), a method for Test-Time Adaptation of Vision-Language Models. M2TPT extends TPT by incorporating three types of "visual memories" according to data previously encountered during the test stream, *i.e.*:

1. A "Multi-Scale Visual Memory" which stores cropped images;
2. A "Holistic Visual Memory" which stores original images (without cropping);
3. An "Irrelevant Memory" that stores cropped images that are deemed wrongly classified;

Given a test image, M2TPT works by:
1. Running standard TPT by computing the marginal entropy over augmented views of the test sample; Following previous works, augmented views are obtained via randomized cropping;
2. Selecting past images from either the "Multi-Scale Visual Memory" or the "Holistic Visual Memory" following the approach of DMN (*i.e.*, images selected from the memory are those belonging to a pseudo-label computed via weighted visual similarity). The choice of the "Multi-Scale Visual Memory" or the "Holistic Visual Memory" depends on which gives the lowest entropy prediction. Once these images are selected, M2TPT computes a cross-entropy objective maximizing the likelihood of the pseudo-label for these selected images;
If the selected memory is the "Multi-Scale Visual Memory", M2TPT retrieves images from the "Irrelevant Memory" as well and adds a KL loss to push these predictions for these images towards a uniform distribution;
3. Updating input prompts by minimizing the three losses above;
3. It updates the "Multi-Scale Visual Memory" by inserting the most confident crop among those that are classified according to the marginal distribution; Insertion at full capacity follows the approach of TDA, *i.e.*, if the memory is full, the new crop is only inserted if there exists at least a previous crop with higher entropy; For the "Holistic Visual Memory" this works analogously, but the source image is inserted instead of a crop;  For the "Irrelevant Memory", the most confident crop is selected among those that disagree with the prediction of the marginal distribution;

The final prediction for a sample is given by averaging (i) the prediction on the source image with the updated prompt and (ii) the prediction based on visual similarity once the memory has been updated.

Experiments are conducted with CLIP-ViT-B/16 on the established evaluation protocol of the field, including Natural Distribution Shifts as well as Finegrained Classification. M2TPT shows state-of-the-art performance compared to the chosen competitors.

**Questions:**

I would start the discussion from the weaknesses above, as well as from the following questions.

1. What are the authors' thoughts about computational complexity? Do they agree/disagree (happy to see disagreement as well!)? Are you open to also reporting runtimes for other competitors and on a "worst-case" scenario like ImageNet?
2. Could you please clarify, in detail, how hyperparameters were selected? I think this is an important aspect for TTA;

Minor comment: This work refers to randomly cropped images as "patches" throughout the whole manuscript. I think this may be very misleading, since the word "patches" usually refers to the tokens of the vision encoder. I would consider using "crops", "augmented views", or similar wording.

**Ethical Concerns:**

["NO or VERY MINOR ethics concerns only"]

**Final Justification:**

The discussion with the authors reached a convergence point, and, to some extent, I agree with (part of) the authors' perspective on complexity vs efficiency. I still fully believe this single paper's comparison point should definitely be expanded to make the paper complete and acceptance-ready. The authors appear willing to broaden their discussion and comparison. However, without clear evidence of this (such as a revised manuscript or a detailed proposal outlining the planned changes), it is difficult for me to envision the final form of the paper, and I am therefore inclined to maintain my "Borderline Reject" rating.

That said, I also acknowledge that this paper has merits, and would not mind it getting accepted.

**Limitations:**

The work contains a limitations section in the Appendix, which looks sufficient. I would suggest moving it to the main body of the paper. I do not think there are evident negative societal impacts of this work.

**Paper Formatting Concerns:**

I do not see any formatting concerns.

**Quality:**

2

**Strengths And Weaknesses:**

**Strengths**.
1. M2TPT shows good performance on both experimental setups (Natural Distribution Shifts and Finegrained Classification);
2. M2TPT can integrate hand-crafted prompt templates in the memory update step, thereby making use of prior knowledge when available;  I think this part also makes evaluation a bit more fair than previous works, as it allows for not giving any advantage to methods assuming hand-crafted templates are available;

**Weaknesses**.
1. I found this paper quite hard to digest; it introduces a convoluted method with many subcomponents, which add a non-negligible complexity. I would suggest moving the pseudocode of the appendix to the main body of the paper;
2. The proposed method carries the computational burden of both single-test-point approaches (e.g. TPT) as well as memory-based ones (e.g. TDA); I do not think Table 4 is a fair representation of the computational burden of M2TPT, for many reasons:
    - The complexity of prompt tuning, as well as of retrieval from the various memories, scales with the number of classes (since backpropagation happens through the layers of the text encoder), and the Table depicts DTD, which is the 2nd-to-first dataset with the smallest number of categories (only EuroSAT has fewer). Probably reporting numbers for ImageNet-1k as well would be beneficial to have a clearer picture;
    - The table compares with other prompt tuning approaches, but other methods are much faster (*e.g.* TDA); I think this work should report runtime and memory usage of the other TTA methods as well;
3. Connecting to the previous point, I think this work is missing a discussion about some important related works on efficient TTA [a, b, c] and, potentially, respective comparisons. I believe this is especially critical since in the past year, the community has observed that simple, optimization-free baselines can work very well for TTA. For example, M2TPT would perform only 0.29% better than [b] on ImageNet and related OOD Benchmarks when using the same templates used in DMN, and I wonder whether this is a sufficient gap to justify the much higher computational complexity added by this work (probably an order of magnitude).
4. The paper contains no explanation about how the hyperparameters of M2TPT were selected;
5. Probably it is amiss not to evaluate with other models than OpenAI's CLIP ViT-B/16. Older papers like TPT experiment with ResNet-50, but I do not think this is super relevant given the current research landscape. In contrast, I would suggest considering different initializations (i.e., sticking to the ViT-B/16 vision encoder, but using OpenCLIP models [d]), and/or scaling to ViT-L/14 as well.


**References**.
[a] Zanella, Maxime, and Ismail Ben Ayed. "On the test-time zero-shot generalization of vision-language models: Do we really need prompt learning?." CVPR 2024.
[b] Farina, Matteo, et al. "Frustratingly Easy Test-Time Adaptation of Vision-Language Models." NeurIPS 2024.
[c] Sui, Elaine, Xiaohan Wang, and Serena Yeung-Levy. "Just shift it: Test-time prototype shifting for zero-shot generalization with vision-language models." WACV 2025.
[d] OpenCLIP repository: https://github.com/mlfoundations/open_clip

---

> ### Author Rebuttal · Authors · 2025-07-31
>
> We thank the reviewer for the constructive comments and suggestions. We appreciate that the reviewer finds our work to have good performance and fair comparisons. Below, we respond to all weaknesses (W) and questions (Q) point by point.
>
> > W1: I found this paper quite hard to digest; it introduces a convoluted method with many subcomponents, which add a non-negligible complexity. I would suggest moving the pseudocode of the appendix to the main body of the paper;
>
> We will follow this suggestion and move the pseudocode to the Method section to improve readability. Moreover, we have revised the overview figure to enhance the correspondence between the figure and the descriptive paragraphs, making the method easier to follow. Overall, our proposed method follows the procedure of visual memory retrieval (patch or holistic) -> prompt tuning with suppression -> memory updates. We hope it makes the overall pipeline more clear.
>
> > W2_1 & Q1: Report computational burden on other datasets, like ImageNet-1k.
> W2_2 & Q1: Report computational burden of backpropagation-free TTA methods apart from test-time prompt tuning methods.
>
> **W2_1 & Q1**. We report the computational burden on ImageNet-1K in the table below. Compared to Table 3, all prompt-tuning methods consume more resources as the number of classes increases due to the increased cost of backpropagation through the text encoder. However, the results still support our claim that M²TPT significantly improves performance with only moderate additional cost—e.g., only 9% additional memory compared to TPT [38].
>
> | Method | Memory (GB) | Runtime (s) | Accuracy (using plain prompt)|
> |--------|--------|---------|---|
> | **Backpropagation-free TTA** |||
> | MTA [a]    |   3.21    |   0.07     | 69.29|
> | DMN [46]   |   2.11    |   0.04     | 69.92|
> | **Test-Time Prompt Tuning** |||
> | TPT [38]   |   17.35    |   0.21     | 68.98|
> | DynaPrompt [42] | >80   |   -     |69.61|
> |M$^2$TPT | 18.96 | 0.30 |71.49|
>
>
> **W2_2 & Q1**. We also report the computational burden of recent backpropagation-free test-time adaptation (TTA) methods. We acknowledge that these methods are computationally cheaper than prompt-tuning-based methods. However, we pursue the prompt-tuning paradigm in this work for two main reasons:
>
> - **Automation and practicality beyond manual prompt engineering**. As shown in Fig. 1(a), prior backpropagation-free methods often outperform TPT on CLIP, largely because they use hand-crafted or LLM-generated prompts. However, such prompts lack automation and scalability, requiring manual engineering for each test scenario and prior knowledge of the test data [38][48]. In contrast, TPT methods enable on-the-fly adaptation to unseen test data. In addition, ***please note that the effort and time consumed for constructing or searching proper and more optimal prompts is always discarded and not included in the table.***
> - **Limitations of previous prompt-tuning methods**. Prior TPT methods rely on limited visual information from the current test image, which restricts the adaptability of the learned prompt. Our method addresses this by enhancing prompt learning with past visual memory, fully exploiting the potential of prompt tuning at test time.
>
> [38] Test-time prompt tuning for zero-shot generalization in vision-language models. NeurIPS2022.\
> [48] Learning to prompt for vision-language models. IJCV2022.
>
> > W3: I think this work is missing a discussion about some important related works on efficient TTA [a, b, c] and, potentially, respective comparisons. I believe this is especially critical since in the past year, the community has observed that simple, optimization-free baselines can work very well for TTA.
>
> To enable a comprehensive comparison, we present results for these methods [a, b, c] on all datasets—with and without prompt engineering (PE) (i.e., hand-crafted and LLM-generated prompts). The table shows that our method consistently outperforms others on average, regardless of whether PE is used.
>
> ***Notably, M$^2$TPT without PE outperforms ZERO + PE, TPS + PE, and MTA + PE***, showing that test-time learned prompts can surpass human-crafted ones. While prompt tuning involves additional computation, it is more practical in real-world scenarios where ***manually engineering prompts for every test domain is infeasible*** [38][48]. As noted in the Introduction, prior prompt-tuning methods fall short compared to prompt-engineering-based methods. Our work advances test-time prompt tuning and demonstrates its ability to outperform strong manually crafted prompt baselines.
>
> | Method | Fine-Grained (Avg of 10) | ImageNet and OOD (Avg of 5) | Avg of 15 Datasets|
> |--------|--------|--------|--------|
> |  MTA [a]   |  64.63 |63.16 | 64.14|
> |  MTA + PE  |  65.58 |64.06 | 65.07|
> |  TPS [c]  |  64.41 |63.03 | 63.95|
> |  TPS + PE  | 66.96 |65.61 |66.51|
> |  ZERO [b]   |  64.21 |63.74 | 64.05|
> | ZERO + PE   | 65.52 |66.10|65.71|
> | M$^2$TPT  | 68.44  | 64.80|67.20 |
> | M$^2$TPT + PE   |**71.34** |**66.39**|**69.69**|
>
> [38] Test-time prompt tuning for zero-shot generalization in vision-language models. NeurIPS2022.\
> [48] Learning to prompt for vision-language models. IJCV2022.
>
> > W4 & Q2: The paper contains no explanation about how the hyperparameters of M2TPT were selected
>
> Following TDA [18], we select hyperparameters using the ImageNet validation set and **fix them across all datasets**. For $\beta$, we choose a relatively small value to maintain a low weight for the irrelevant suppression loss when the number of classes is small. We also include a sensitivity analysis on 10 fine-grained datasets, shown below. The performance varies only slightly, indicating that our method is not highly sensitive to $\beta$.
> |  | $\beta=0.05$ | $\beta=0.1$ | $\beta=0.15$| $\beta=0.20$|
> |--------|-----------|-----------|-----------|-------|
> |Avg accuracy | 68.46 | 68.44 | 68.26 | 68.23|
>
>
> > W5: I would suggest considering different initializations (i.e., sticking to the ViT-B/16 vision encoder, but using OpenCLIP models [d]), and/or scaling to ViT-L/14 as well.
>
> We follow your suggestion and scale our evaluation from ViT-B/16 to ViT-L/14 using the **plain prompt** and unchanged hyperparameters. The results below show that M$^2$TPT achieves a 2.77% average improvement over other methods across 10 datasets, indicating robust performance gains across model scales. We will include it in the revision.
>
> | Method | Flower | DTD | Pets | Cars | UCF | Caltech | Food | SUN | Aircraft | EuroSAT | Average |
> |--------|--------|-----|------|------|-----|---------|------|-----|----------|---------|---------|
> |  TPT   |  76.49 |53.55| 93.57| 77.96|75.10| **95.21**   | 89.43|69.40| 31.68    | 47.56   | 71.00   |
> | ZERO   | 76.41  |53.63|94.08 |78.39 |74.68|**95.21**    |90.66 |69.61| **33.62**    | 44.21   |71.05    |
> |**M$^2$TPT**| **77.43**  |**56.91**|**94.79** |**79.23** |**79.57**|94.28    |**91.71** |**71.04**| 32.01    | **61.20**   |**73.82**    |

---

> ### Comment · Reviewer_D3JX · 2025-08-04
>
> Dear Authors,
>
> Thank you for the detailed rebuttal! Let me reply below:
>
> **Computational complexity of M2TPT.** Thank you for reporting these numbers! Unfortunately, as anticipated, what puzzles me is the increased computational cost of M2TPT compared to recent efforts in TTA. According to the numbers above, compared to recent efforts such as DMN and MTA, the proposed solution increases both space and time requirements by significant margins ($4.3\times$ and $7.5\times$ slower than MTA and DMN, respectively, while consuming $5.9\times$ and $9\times$ more memory). I understand the merits of increased accuracy, but the computational requirements look a bit too excessive to me. I am aware that "the effort and time consumed for constructing or searching proper and more optimal prompts is always discarded and not included in the table" when it comes to related works, but I believe this is for a valid reason, *i.e.*, such a cost is amortized after deployment (if you think of it, for research benchmarks this was done once in the CLIP paper and never again in a multitude of follow-up works). This is not the case if inference on a single test-point is significantly more demanding, as the gap in compute spent will constantly increase over time.
>
> **M2TPT vs efficient TTA baselines.** Thank you for reporting these numbers! For the "Fine-Grained (Avg of 10)" column, it looks like these results were directly reported from the provided references, both with and without PE. I guess there's a big caveat here, which is that both [a] and [b] (MTA and ZERO) do not use dataset-specific templates, but rather just ensemble a set of generic prompts provided by the CLIP authors for ImageNet alone. In contrast, this paper claims to use the same templates of DMN, which are dataset-specific. This is, unfortunately, not a fair comparison and gives an advantage to the method proposed in this paper. I think it would be beneficial to examine such a comparison.
>
> About all other points, I have no further comment, and I appreciate your responses. The two aforementioned points are, unfortunately, what make me refrain from switching from a reject to an accept score at the current stage.
>
> All the best,
> Reviewer `D3JX`

---

> > ### Author Response · Authors · 2025-08-05
> > **Response to Reviewer D3JX**
> >
> > We sincerely thank Reviewer D3JX for carefully reading our rebuttal and providing thoughtful follow-up comments. We would like to offer additional details and share our perspective in the open discussion:
> >
> > We would like to emphasize that a direct comparison of efficiency between training-free and prompt tuning-based approaches—two fundamentally different categories of TTA methods—may not provide a fully fair or meaningful assessment. By design, prompt tuning-based methods inevitably incur additional computational overhead, making them inherently more resource-intensive. Therefore, we believe it is more reasonable and informative to compare efficiency within each methodological category, rather than across them.
> >
> > If computational efficiency were to be treated as the sole metric for evaluating TTA effectiveness, then by that logic, all optimization-based methods, including prompt tuning, would be completely disqualified. However, this is clearly not reflected in the research trajectory of the community. For instance, the publication of DMN (CVPR'24), a training-free method, did not curtail the development or acceptance of optimization-based methods, i.e, DynaPrompt was published at ICLR'25 following DMN, demonstrating continued interest and progress in this line of work. As reported in the rebuttal Table, DynaPrompt still demands significantly more resources compared to our M2TPT, further highlighting the value of efficient yet effective methods like ours within the optimization-based category.
> >
> > A similar dynamic exists in other fields e.g., 3D reconstruction, where both feedforward and optimization-based methods coexist and advance in parallel. While feedforward approaches do not require scene-specific optimization and are generally more efficient, optimization-based methods continue to thrive due to their distinct advantages in quality and flexibility. Despite the efficiency of feedforward methods, optimization-based approaches, such as NeRF[A] and 3DGS[B], remain central and highly active areas of research. The coexistence of these directions has been mutually beneficial and has enriched the field as a whole.
> >
> > We hope that the TTA community can similarly support the development of both training-free and optimization-based methods. We believe that encouraging diverse research directions will foster innovation and contribute to a more comprehensive understanding of the TTA problem space.
> >
> > [A]NeRF: Representing Scenes as Neural Radiance Fields for View Synthesis,ECCV'20.
> >
> > [B]3D Gaussian Splatting for Real-Time Radiance Field Rendering,SIGGRAPH'23.
> >
> > > ...such a cost is amortized after deployment (... this was done once in the CLIP paper and never again in a multitude of follow-up works)
> >
> > Additionally, we would like to clarify that recent SoTA training-free TTA methods rely not only on the 80 prompts from the CLIP paper, but also on hand-crafted and LLM-generated prompts specific to each test dataset, as seen in DMN, TDA, and AWT. The “**Automation and practicality beyond manual prompt engineering**” in the initial response is based on the fact that such dataset-specific prompt engineering requires ***prior knowledge of the test data, substantial engineering effort and time, and domain expertise—all of which present challenges for deployment***. By contrast, test-time prompt tuning can adapt to the test data on the fly, demonstrating prominent practicality from this perspective.
> >
> > > **M2TPT vs efficient TTA baselines**....MTA and ZERO do not use dataset-specific templates but a set of generic prompts provided by the CLIP authors...it would be beneficial to examine such a comparison.
> >
> > We thank the reviewer for clarifying the details. Accordingly, to ensure a fair comparison, we add results for MTA and ZERO using the dataset-specific prompts from DMN, and update the table below. Consistent with our previous conclusion, our method outperforms others on average in both “with” and “without” prompt engineering. Moreover, the further improvements of MTA and ZERO when using heavier LLM-generated prompts support our argument that the superior performance of recent training-free methods over recent test-time prompt tuning methods largely benefits from prompt engineering.
> >
> > Importantly, our M$^2$TPT ***without*** PE is comparable to MTA+PE(DMN) and ZERO+PE(DMN). This suggests that the prompts learned at test time from test data streams on the fly in our method can achieve an ability comparable to that of powerful LLM-generated prompts, demonstrating that our work addresses the bottleneck of the TPT category of methods and outperforms SoTA.
> >
> > |Method|Fine-Grained|ImageNet+OOD|Avg of 15 Datasets|
> > |-|-|-|-|
> > |MTA|64.63|63.16|64.14|
> > |MTA+PE(CLIP)|65.58|64.06|65.07|
> > |MTA+PE(DMN)|69.16|64.06|67.46|
> > |TPS|64.41|63.03|63.95|
> > |TPS+PE|66.96|65.61|66.51|
> > |ZERO|64.21|63.74|64.05|
> > |ZERO+PE(CLIP)|65.52|66.10|65.71|
> > |ZERO+PE(DMN)|68.74|66.10|67.86|
> > |M$^2$TPT|68.44|64.80|67.23|
> > |M$^2$TPT+PE|**71.34**|**66.39**|**69.69**|

---

> > > ### Comment · Reviewer_D3JX · 2025-08-05
> > >
> > > Dear Authors,
> > >
> > > Thank you for sharing your perspective!
> > >
> > > **Efficient vs Optimization-based TTA**. I understand your point, and I think I agree to some extent. I especially agree with the following sentence: *"if computational efficiency were to be treated as the sole metric for evaluating TTA effectiveness, then by that logic, all optimization-based methods, including prompt tuning, would be completely disqualified."* There's also a part I do not agree with, though, which is: *"we believe it is more reasonable and informative to compare efficiency within each methodological category, rather than across them."* In general, when it comes to any field, although there may be subcategories, I think it's beneficial for readers to know that there are alternatives to the proposed line of work, which might have both pros and cons. Including comparisons across different lines of work provides a comprehensive view and makes the paper inherently more complete, in my opinion. I also think that such a full picture allows for a better understanding of the merits of a paper, *i.e.*, although I do not think this applies to this submission, there might not be much merit in a paper that achieves minimal performance improvement over much lighter alternatives.
> > > So, a follow-up question from my side is: are you open to reporting such comparisons in the main paper, both in terms of efficiency and raw performance? If not, why so?
> > >
> > > **M2TPT vs Efficient TTA w/ equal PE.** Thank you for reporting these as well! I think the response is sufficient, and I do not have any further comments here.
> > >
> > > Happy to discuss this further,
> > > Reviewer `D3JX`

---

> > > > ### Author Response · Authors · 2025-08-07
> > > > **Thank You for Your Continued Engagement**
> > > >
> > > > Dear Reviewer D3JX,
> > > >
> > > > Thank you for your valuable review and continued feedback during the discussion phase. We understand your concerns and questions throughout the reviews and follow-up comments, including: (1) experiments with different initializations, (2) clarification on hyperparameter selection, (3) a more comprehensive comparison with training-free TTA methods (e.g., ZERO and MTA), and (4) the higher computational burden of test-time prompt tuning compared to training-free TTA.
> > > >
> > > > We appreciate that you acknowledged our responses to points (1)–(3), including the added experiments on larger CLIP models, the clarification of hyperparameter settings, and the detailed comparison with training-free TTA methods.
> > > >
> > > > Regarding the concern on computational efficiency (point 4), we truly appreciated the opportunity to engage in a thoughtful discussion on test-time prompt tuning and training-free TTA as two lines of work within the field. While prompt tuning is inherently more resource-intensive, we have provided detailed reasoning and empirical comparisons to highlight its practical value—for example, it eliminates the need for extensive prompt engineering and achieves significant accuracy improvements with our M$^2$TPT method. These results help clarify the trade-offs involved and demonstrate the viability of prompt tuning in real-world deployment scenarios.
> > > >
> > > > We’d like to kindly check if you have any further questions or comments—we’d be happy to continue the discussion with you before the phase concludes.
> > > >
> > > > Best regards,\
> > > > The Authors

---

> ### Author Response · Authors · 2025-08-05
> **Response to Reviewer D3JX**
>
> We sincerely thank the reviewer for the continued engagement and thoughtful feedback. We greatly value the reviewer’s perspective on the importance of presenting a comprehensive view of the field, including alternatives with their respective strengths and limitations. Although this paper focuses on the prompt-tuning line of work, we have already included substantial discussion of the training-free TTA line in the Introduction, Related Work, and Experiments sections. In addition to the existing discussion, we will also provide a detailed comparison and discussion of both efficiency and accuracy between the two lines of work in the main paper.
>
> We would also like to clarify that the performance improvement achieved by our work is not minimal but **significant**. Specifically, compared to MTA and ZERO, M$^2$TPT shows a **3.18**% average accuracy improvement across 15 datasets without PE. Compared to the most recent SoTA test-time prompt tuning method (DynaPrompt), M$^2$TPT without PE achieves a **2.43**% average improvement across the same 15 datasets, while also requiring much less computational overhead.
>
> We agree that both accuracy and efficiency are important across all lines of work, and it is expected that the community will continue to advance on both fronts. Ultimately, it is the end user who decides which method to use—depending on the application, even a 1% accuracy improvement might justify additional computational cost. Our method primarily focuses on improving accuracy, with less emphasis on efficiency. As future work, we aim to better integrate training-free and optimization-based approaches, fostering their joint development to achieve a more balanced and practical trade-off between accuracy and efficiency.

---

### Official Review · Reviewer_CxW2 · 2025-07-01

**Clarity:** 2
**Significance:** 3
**Originality:** 3
**Rating:** 4
**Confidence:** 4

**Summary:**

This paper proposes M²TPT, a test-time prompt tuning method for CLIP that incorporates a multi-scale visual memory to provide richer class-relevant information during prompt optimization. To address the limitations of relying on a single test image, the method retrieves and updates visual memory in a mutual feedback loop. Two key improvements are introduced: a holistic visual memory to better support image-level tasks, and an irrelevance suppression strategy that filters noisy memory and penalizes misleading high-confidence patches.Experiments show that the proposed method significantly improves performance over existing TPT baselines.

**Questions:**

1. The paper proposes using two independent memory banks (multi-scale and holistic), but during inference, only one of them is selected for prompt tuning based on entropy. Given that both local and global visual information could be beneficial for prompt learning, why does the method ultimately choose to use only one memory type rather than integrating both?

2. In the comparison with prompt-tuning-based methods, the proposed method still underperforms on ImageNet-V2 and ImageNet-R. Could the authors provide insights into the potential causes of this gap, and whether there are ways to further improve performance on these challenging datasets?

**Ethical Concerns:**

["NO or VERY MINOR ethics concerns only"]

**Final Justification:**

Most of my concerns have been resolved and I have decided to keep my original score.

**Limitations:**

Yes.

**Quality:**

3

**Strengths And Weaknesses:**

Strengths

1. The method is mathematically well-grounded, with rigorous logic and solid theoretical support.

2. It innovatively extends traditional memory-based frameworks by introducing two separate memory modules (multi-scale and holistic) along with a noise suppression mechanism, improving robustness.

Weaknesses

1. The overall pipeline figure is overly complex and lacks clarity, making it hard to follow the method at a glance.

2. The descriptions of the two main improvements (holistic memory and irrelevance suppression) are too brief, making their motivations and contributions difficult to fully grasp.

---

> ### Author Rebuttal · Authors · 2025-07-31
>
> We thank the reviewer for the constructive comments and suggestions. We appreciate that the reviewer finds our method novel and mathematically well-grounded, with rigorous logic. Below, we respond to all weaknesses (W) and questions (Q) point by point.
>
> > W1: The overall pipeline figure is overly complex and lacks clarity, making it hard to follow the method at a glance.
>
> We have improved the overview figure by making the three components—memory retrieval, prompt tuning, and memory update—more modular and visually distinct. Additionally, we enhanced the correspondence between the illustration and method descriptions by adding equation indices to the figure, thereby improving readability.
>
> > W2: The descriptions of the two main improvements (holistic memory and irrelevance suppression) are too brief, making their motivations and contributions difficult to fully grasp.
>
> We first introduce the motivations for these two components in detail in lines 57–67 of the Introduction. These motivations are then recalled and reinforced in the Method section—specifically in lines 182–186 and lines 198–201. We would like to add detailed descriptions as:
>
> The holistic memory complements the multi-scale memory, particularly for holistic visual recognition tasks such as scene classification. These tasks require global visual descriptions, which may be overlooked by using only multi-scale memory. To address this, we design the holistic memory to provide comprehensive image-level context for each class during memory retrieval.
>
> The irrelevance suppression component aims to mitigate the adverse effects of high-confidence mispredictions during test time. Since the memory retrieval and update processes rely on test-time predictions, the visual memory is inevitably noisy due to prediction errors. To reduce the impact of this noise, we first filter out low-relevance memory entries during retrieval. Furthermore, we construct a class-irrelevant memory to store the most “conspicuous” noise—i.e., high-confidence but incorrect patches—and suppress their influence during prompt tuning.
>
> Moreover, we will add a more detailed mathematical formulation of the holistic memory in the Appendix. Additionally, we include a pseudocode of the full pipeline in Appendix A, which we will move to the main body to enhance clarity and accessibility.
>
>
> > Q1: Why does the method ultimately choose to use only one memory from multi-scale and holistic memory rather than integrating both?
>
> The use of holistic memory is motivated by tasks requiring global visual understanding. In such tasks, relying on local patches (as in multi-scale memory) may lead to low-confidence or distracting cues. For instance, in scene classification which relies on global visual description, an isolated object patch may introduce confusion due to its limited local context. Therefore, in these cases, our method selects the high-confidence holistic memory and disregards the multi-scale one—and vice versa—depending on the entropy-based confidence measure.
>
> > Q2: In the comparison with prompt-tuning-based methods, the proposed method still underperforms on ImageNet-V2 and ImageNet-R. Could the authors provide insights into the potential causes of this gap, and whether there are ways to further improve performance on these challenging datasets?
>
> We hypothesize that the performance gap arises from the high intra-class variation in datasets like ImageNet-R. Such variation can adversely affect the memory retrieval process, which relies on visual similarity. To mitigate this issue, future work could explore strategies to explicitly disentangle intra-class variation from inter-class differences, enhancing the robustness of memory retrieval.

---

> > ### Comment · Reviewer_CxW2 · 2025-08-06
> > **response to rebuttal**
> >
> > Thanks to the author's response, most of my concerns have been resolved and I have decided to keep my original score.

---

> ### Author Response · Authors · 2025-08-05
> **Thank You and Gentle Reminder**
>
> Dear Reviewer CxW2,
>
> We sincerely appreciate your valuable feedback and the time you have taken to review our work. As the discussion period will end within 48 hours, we just wanted to kindly check if you might have any further questions or points you would like to discuss.
>
> We would be very happy to provide additional clarification or details should you need them.
>
> Best regards,\
> The Authors

---

> ### Author Response · Authors · 2025-08-07
> **Thank you for your response**
>
> Dear Reviewer CxW2,
>
> Thank you for your response and for your thoughtful feedback throughout the review process. We're glad to hear that our rebuttal addressed your concerns and deeply appreciate that you remain positive on our submission. If you have any further questions or thoughts, we’d be happy to discuss them before the end of the discussion phase.
>
> Best regards,
>
> The Authors

---

### Note · Authors · 2025-08-12

We sincerely thank the ACs, SACs, and PCs for handling our submission, and we greatly appreciate all reviewers for their time and effort in evaluating our paper.

Our work initially received recognition from the reviewers for its novelty [CxW2, rbTk], good performance [D3JX, e7b1], and detailed experiments [rbTk]. Reviewers CxW2, rbTk, and e7b1 subsequently acknowledged that our rebuttal addressed their concerns and each gave a rating of 4.

We are especially grateful to reviewer D3JX for the multi-round discussion, during which we engaged deeply on the reviewer’s concerns regarding: (1) scalability to larger CLIP models, (2) hyperparameter selection, (3) comparison with training-free TTA works, and (4) the higher computational cost of prompt tuning compared to training-free TTA. The reviewer explicitly acknowledged that points (1)–(3) were sufficiently addressed. For point (4), we view this as part of a broader, ongoing discussion in the TTA community regarding the trade-offs between prompt tuning and training-free approaches, rather than a question specific to our work. In our response, we provided detailed reasoning and empirical evidence demonstrating the practical value of prompt tuning. Its flexibility—eliminating the need for extensive prompt engineering—and the substantial accuracy gains shown by our M$^2$TPT method highlight its viability for real-world deployment and its importance as an active, evolving research direction.

We sincerely thank all reviewers for their thoughtful and constructive feedback, which has helped strengthen our paper.

---

### Decision · Program_Chairs · 2025-09-17

**Decision:**

Reject

**Comment:**

This work studies test-time prompt tuning (TPT), and attributes the performance gap of existing methods to limited class-specific visual knowledge derived from a single test image. To tackle this challenge, this work proposed multi-scale visual memory to store class-relevant image patches, global image-level context, and noisy memory reduction. Experiments conducted on 15 benchmarks validate the effectiveness of the proposed method compared to existing TPT methods.

The main strengths are that (1) the multi-scale visual memory module is a novel extension and improves the robustness, (2) the empical performance is strong accross different test-time adaptation setups, (3) the ablation studies are thorough and validate the contribution of each component.

The main weaknesses are that (1) the method with multiple components is complex and not easy to follow, (2) the writing quality w.r.t. clarity and motivation of key components, (3) robustness issues with test input ordering, misclassified samples in the irrelevance suppression stage, and no long-term memory management strategy.

During the rebuttal and discussion, most of the concerns are addressed, except for the evidence of improving writing quality in aspects of discussion and comparison, indicating the paper needs more than a minor revision for publication. Besides, the method is a bit complex consisting of multiple components, leading to a less general approach to the broader field. Overall, this paper is a borderline paper and needs further polishment. Considering the overall quality of this paper and other papers in the batch, AC decided not to recommend accepting this paper at this stage.